# Evaluation of a Lecithin Supplementation on Growth Performance, Meat Quality, Lipid Metabolism, and Cecum Microbiota of Broilers

**DOI:** 10.3390/ani11092537

**Published:** 2021-08-29

**Authors:** Yiru Shen, Shan Zhang, Xu Zhao, Shourong Shi

**Affiliations:** 1Poultry Institute, Chinese Academy of Agricultural Sciences, Yangzhou 225125, China; shenyiru929@126.com (Y.S.); zhangshan3321@163.com (S.Z.); kity850814@163.com (X.Z.); 2Institute of Effective Evaluation of Feed and Feed Additive (Poultry Institute), Ministry of Agriculture, Yangzhou 225125, China; 3Jiangsu Co-Innovation Center for Prevention and Control of Important Animal Infectious Diseases and Zoonoses, Yangzhou 225000, China

**Keywords:** lecithin, broilers, performance, lipid metabolism, gut microbial community

## Abstract

**Simple Summary:**

Lecithin can not only provide energy to animals but also serves as an emulsifier and has the potential to enhance the utilization of dietary fat by animals. Thus, there is a need to elucidate the underlying mechanism of the positive effect in broilers. The present feeding trial aims to evaluate the effect of lecithin on broilers’ performance, meat quality, lipid metabolism, and cecum microbiota. The obtained results revealed significant improvements in broiler meat quality resulting from the lipid metabolism and microbiota that were affected by lecithin treatment. Consequently, it could be used in broilers’ diets for the aim of meat quality improvement.

**Abstract:**

The present study was conducted to evaluate the effects of lecithin on the performance, meat quality, lipid metabolism, and cecum microbiota of broilers. One hundred and ninety-two one-day-old AA broilers with similar body weights (38 ± 1.0 g) were randomly assigned to two groups with six replicates of sixteen birds each and were supplemented with 0 and 1 g/kg of lecithin for forty-two days. Performance and clinical observations were measured and recorded throughout the study. Relative organ weight, meat quality, lipid-related biochemical parameters and enzyme activities were also measured. Compared with broilers in the control group, broilers in the lecithin treatment group showed a significant increase in L* value and tenderness (*p* < 0.05). Meanwhile, the abdominal adipose index of broilers was markedly decreased in lecithin treatment after 42 days (*p* < 0.05). In the lipid metabolism, broilers in the lecithin treatment group showed a significant increase in hepatic lipase and general esterase values at 21 days compared with the control group (*p* < 0.05). Lower *Firmicutes* and higher *Bacteroidetes* levels in phylum levels were observed in the lecithin treatment group after 21 and 42 days. The distribution of *lactobacillus*, *clostridia*, and *rikenella* in genus levels were higher in the lecithin treatment group after 21 and 42 days. No statistically significant changes were observed in performance, relative organ weight, or other serum parameters (*p* > 0.05). These results indicate that supplementation with lecithin significantly influence the lipid metabolism in broilers at 21 and 42 days, which resulted in the positive effect on the meat color, tenderness, and abdominal adipose in broilers.

## 1. Introduction

Phospholipids, which are found in thousands of organisms, are key components of the cell membrane. Dietary intake of exogenous phospholipids provides the majority of phospholipids organisms need. Lecithin, which is mostly obtained from soybean, oilseed rape, and sunflower seed in a normal diet, is widely known to be an important transporter of lipids in organisms [1]. Commercially available products are mostly extracted from soybeans, and they are widely used in healthcare for humans and animals [2]. Lecithin can not only provide energy to animals but also serves as an emulsifier and has the potential to enhance utilization of dietary fat by animals [3]. The physiological effects of lecithin have been studied extensively in recent years, although public health recommendations regarding lecithin intake currently have no limit. Even so, lecithin has become a popular animal dietary supplement for increasing performance and nutrient utilization. The effect of lecithin on cholesterol reduction was validated in monkeys, hamsters, and many other species [4]. It was reported in a previous study that a diet supplemented with lecithin could increase the daily gain of nutrients and nutrient digestibility in animals [5]. Moreover, lecithin treatment in intestinal cell membranes alters the permeability of cell bilayers and resulted in the greater influx of micro- and macro-molecules across the cell membrane [6]. The positive effect of lecithin is also attributed to healthy gut improvement [7]. Therefore, supplementing exogenous lecithin emulsifiers in the diets has become very popular in poultry production.

To date, limited studies have been well conducted with exogenous lecithin or emulsifiers, and inconsistent responses in broilers have been noted. Only a few studies on broilers reported that lecithin can maintain broiler performance with low energy diets [8]. Several studies reported that supplementation with lecithin to the diet of broilers improved growth performance [9]. Meanwhile, many studies of lecithin treatment in broilers show the positive effect on the improvement of apparent energy and nutrient utilization [10]. Contrarily, it has been reported that emulsifiers have no significant impact on the growth performance of broilers [11,12]. The different effect of lecithin or emulsifiers on broilers are attributed to the ingredients and test conditions. However, these studies have barely noticed the effect of lecithin on the gut microbial community of broilers. As the pivotal component of intestinal barrier, the composition and function of the gut microbiota is dynamic and affected by diet properties. Meanwhile, the gut microbiota has shown effect on lipid metabolism and lipid levels in blood and tissues [13]. Hence, we hypothesized that lecithin supplementation might have unknown effects on microbial communities in broilers, which might further affect the lipid digestibility and utilization in broilers. Therefore, the present study was conducted to evaluate the effects of lecithin on performance, meat quality, lipid metabolism, and the microbial community of broilers.

## 2. Materials and Methods

### 2.1. Birds, Diets, and Management

A total of one hundred and ninety-two one-day male Arbor Acres (AA) broilers were randomly assigned to two groups (control and treatment) with six replicates of sixteen birds each. The birds of each replicate were reared in a single cage (2.4 × 0.6 × 0.6 m) with a wire screen floor. Water and feed were provided ad libitum, with the photoperiod set at 23 L:1 D throughout the study. The temperature in the broiler house during the first week was 32 to 35 °C, after which it was lowered by 1 °C every other day until it reached 27 °C. The study was conducted according to the Regulations of the Experimental Animal Administration issued by the State Committee of Science and Technology of the People’s Republic of China. The animal use protocol was approved by the Animal Care and Use Committee of the Poultry Institute at the Chinese Academy of Agriculture Science.

Lecithin was derived from soybeans (PHOSPHOLIPON 90 g, Lipoid Co. Ltd., Ludwigshafen, Germany) with 94.3% purity, 1.3% nonpolar lipids, 1.2% lysophosphatidylcholine, 0.2% water, 0.17% tocopherol, and other sterols. The diet of the birds was formulated to meet or slightly exceed all nutrient requirements (Table 1) (NRC, 1994) [14], and it was provided in mash form to avoid degradation of lecithin. All of the birds were fed diets supplemented with 0 (control) and 1 g/kg (treatment) lecithin for 42 days.

### 2.2. Growth Performance and Sample Collection

Cage-side observations, which included recording changes in clinical condition or behavior, were made at least twice daily throughout the study. All macroscopic abnormalities in the birds or deaths throughout the whole experiment were recorded after necropsy. The body weights of the birds from different replicate were determined at the beginning, 21 days, and 42 days into the study. Feed consumption was recorded on a replicate basis at 21 and 42 days. Feed conversion was expressed as the grams of feed consumed per grams of weight gain. The average daily gain (ADG), average daily intake (ADI) and FCR were calculated at 1 to 21 days of age, 22 to 42 days of age, and 1 to 42 days of age.

At days 21 and 42, 1 bird was randomly selected from each replicate (6 birds from each group) and weighed after 8 h of feed deprivation. Before the necropsy, 1.5 mL wing blood were centrifuged at 3500 g for 10 min so that serum could be collected for clinical blood chemistry and enzyme activity detection. After that, the birds were fed for 4 h and sacrificed by jugular bleeding, while the organs (liver, heart, spleen, and thymus) were removed and weighed. The abdominal adipose tissue of broilers were removed and weighed at 42 days. Breast muscle (both the pectoralis major and minor included) was collected from the right pectorals and stored at 4 °C for later analysis. The contents from both ceca were thoroughly mixed and stored at −80 °C for 16S rDNA amplicon sequencing analysis.

### 2.3. Meat Quality

The meat samples were stored in 4 °C for 24 h before the detection. Indices of meat quality including pH, color, shear force, and drip loss were determined with the methods described previously [15,16].

The pH value was measured with a portable pH meter (HI8424, Beijing Hanna Instruments Science & Technology Co. Ltd., Beijing, China) equipped with an insertion glass electrode calibrated in buffers at pH 4.01 and 7.00 at ambient temperatures. The measurements were made at the same location on individual breast and thigh muscle samples. The average pH value was calculated from 3 readings taken on the same muscle sample.

Meat color was assessed with a chroma meter (CR-10, Minolta Co. Ltd., Suitashi, Osaka, Japan) to measure CIE LAB values (L* means relative lightness, a* means relative redness, and b* means relative yellowness). The tip of the colorimeter measuring head was placed flat against the surface of the muscle. The meat color was expressed using the CIELAB dimensions of lightness (L), redness (a), and yellowness (b). The higher L* values were lighter, higher a* values were more red, and higher b* values were more yellow. Drip loss was estimated by determining expressible juice using a modification of the filter paper press method. A raw meat sample weighing 1.0 g was placed between 18 pieces of 11 cm diameter filter paper and pressed at 35 kg for 5 min at 25 °C. The expressed fluid was determined as the change in the weight of the original sample. The water-holding capacity was calculated as the ratio of expressible fluid/total moisture content.

A shear force test was done on the breast fillets using the razor blade method with an Instron Universal Mechanical Machine (Instron model 4411, Instron Crop., Canton, MA, USA). Meat samples were stored at 4 °C for 24 h and were then individually cooked in a water bath at 80 °C in plastic bags to an internal temperature of 70 °C. The samples were then removed and chilled to room temperature. Strips (1.0 cm (width) × 0.5 cm (thickness) × 2.5 cm (length)) parallel to the muscle fibers were prepared from the medial portion of the muscle and sheared vertically. Shear force was expressed in kilograms. Three values were recorded for each replicate sample and averaged.

### 2.4. Lipid Metabolism

Biochemical parameter detection was performed in serum immediately. Glucose (GLU), cholesterol (CHO), triglyceride (TG), low-density lipoprotein cholesterol (LDL), and high-density lipoprotein cholesterol (HDL) were measured using an Unicel DXC 800 (Beckman Coulter, Fullerton, CA, USA).

Lipoprotein lipase (LPL) and hepatic lipase (HL) in serum were measured with colorimetric enzymatic methods using commercially available kits (Nanjing Jiancheng Bioengineering Institute, Nanjing, Jiangsu, China). General esterase (GE) activity was calculated as the sum of LPL and HL.

HL detection in liver was performed in liver supernatant. Approximately 0.1 g of liver sample was transferred into a 1.5 mL precooled centrifuge tube with 0.9 mL physiologic saline and homogenized into 10% homogenates (50 Hz for 3 min through tissue grinder, SCIENZT-48, Scientz Biotechnology Co., Ltd., Ningbo, Zhejiang Province, China). It was then centrifuged through a low-speed centrifuge (2500–3000 rpm/min for 10 min, DL-5M, Xiangyi Power Testing Instrument Co. Ltd., Changsha, Hunan, China). The test kit was the same as serum.

### 2.5. DNA Extraction, PCR Amplification of 16S rDNA, Amplicon Sequence, and Sequence Data Processing

Microbial genomic DNA was extracted from 220 mg of cecal contents sample using a QIAamp DNA Stool Mini Kit (Tiangen Biotech Company Limited, Beijing, China) following the manufacturer’s instructions. Successful DNA isolation was confirmed by an A260/280 ratio ranging between 1.8 and 2.0 and by agarose gel electrophoresis.

Based on previous comparisons, the V4 hypervariable regions of 16S rDNA were PCR amplified from microbial genomic DNA harvested from samples and were used for the remainder of the study. PCR primers flanking the V4 hypervariable region of bacterial 16S rDNA were designed. The barcoded fusion forward primer was 520F 5′-barcode + GCACCTAAYTGGGYDTAAAGNG-3′, and the reverse primer was 802R 5′-TACNVGGGTATCTAATCC-3′. The PCR conditions were as follows: one pre-denaturation cycle at 98 °C for 30 s, 25 cycles of denaturation at 98 °C for 15 s, annealing at 50 °C for 30 s, and elongation at 72 °C for 30 s, and one post-elongation cycle at 72 °C for 5 min. The PCR amplicon products were separated on 0.8% agarose gels and extracted from the gels. Only PCR products without primer dimers and contaminant bands were collected for sequencing by synthesis (Axygen Axy Prep DNA Gel Extraction kit, New York, NY, USA). Barcoded V4 amplicons were sequenced using the paired-end method by Illumina MiSeq (Sangon Biotech Company Limited, Shanghai, China) with a 600-cycle index read. Only sequences with an overlap longer than 10 bp and without any mismatch were assembled according to their overlap sequence. Reads that could not be assembled were discarded. Barcode and sequencing primers were trimmed from the assembled sequence [17,18].

### 2.6. Statistical Analysis

In this study, operational taxonomic unit (OTU) cluster analysis was used to classify the OTU sequences based on a 97% similarity criterion. The OTU abundance of each sample was generated at the genus level. The bacterial diversity is shown by the number of OTUs. The mean length of all effective bacterial sequences without primers was 280 bp. The abundance and diversity of microbiota were compared between each sample by calculating OTUs.

A pen of birds was the experimental unit for performance parameters. For all other measurements we used individual birds from each replicate. All the data are presented as the means ± SEM. Statistical analyses were carried out with SPSS 18.0 for windows (SPSS Inc., Chicago, IL, USA). Differences between groups were tested with a *t*-test for independent samples. A *p* value less than 0.05 was considered to indicate statistical significance, and a trend was considered present at *p* < 0.10.

## 3. Results

### 3.1. Performance

The mean performance (ADG, ADI, and FCR) and mortality are shown in Table 2. No statistically significant differences were found among all performance parameters or mortality observed in either the treated group or the control group throughout the whole period (*p* > 0.05). The relative organ weight results are shown in Table 3. As shown in Table 3, no statistically significant changes were observed in relative organ weight of broilers in 21 or 42 days (*p* > 0.05). The abdominal adipose tissue of the control group at 42 days was significantly higher compared with the lecithin treatment groups (*p* < 0.05).

### 3.2. Meat Quality

The meat quality parameters results can be seen in Table 4. The L values were markedly higher in the lecithin treatment groups compared with the control group (*p* < 0.05). Drip loss and b value were higher in the lecithin treatment group than in the control groups, while the *p* values were 0.062 and 0.078 separately. the tenderness value (shear force) was markedly lower in the lecithin treatment groups than in the control group (*p* < 0.05) at 42 days. No other parameters were significantly affected by dietary lecithin supplementation (*p* > 0.05) at 21 or 42 days.

### 3.3. Lipid Metabolism

The results of the lipid-related biochemical parameters in serum are shown in Table 5, while the results of lipid-related enzyme activity in serum are shown in Table 6. As shown in Table 5, the GLU value at 42 days was significantly higher in lecithin-supplemented groups than in the control groups (*p* < 0.05), which confirmed that exogenous high lipid intake would increase the glucose metabolism in vivo. Meanwhile, the cholesterol level in serum of broilers at 21 days significantly decreased with lecithin supplementation, but no difference at 42 days.

### 3.4. 16S rDNA Analysis of Bacterial Communities

Shifts of cecal microbial community along with body development: The results shown in Figure 1 and Figure 2 describe the distribution of DNA sequences into phyla after 21 and 42 days. For the majority of phylum in intestinal content, Firmicutes was the most dominant phylum for all development stages. However, the abundance of Firmicutes (88.10%) in the control group was significantly higher than that in the lecithin treatment group (78.83%) after 21 days. Conversely, the *Bacteroidetes* level in the control group (5.34%) was significantly lower than that in the lecithin treatment groups (11.99%) after 21 days. *Proteobacteria* were observed at a higher level in lecithin treatment groups (11.99%) after 21 days. The results after 42 days were the same as after 21 days. The abundance of *Firmicutes* (82.26%) in the control group was significantly higher than that in the lecithin treatment group (70.10%) after 42 days. Conversely, the *Bacteroidetes* level in the control group (12.39%) was significantly lower than that in the lecithin treatment groups (26.19%) after 42 days.

Principal components analysis (PCA) plots of samples from after 21 (A) and 42 (B) days: PCA results (Figure 3) illustrated the differences in distribution of microbial community. Based on the PCA analysis, the control group and the lecithin-supplemented group were significantly divided into two clusters. Combined with previous data, this result suggested that the lecithin treatment group had a significantly different gut micro-flora profile.

Genus level significance analysis of the cecal microbial community after 21 and 42 days: Table 7 and Table 8 represent the abundance of selected genera (>0.1% in at least one sample) across all samples. It clearly showed that there were apparent differences in genus distribution between control and lecithin treatment groups. The proportions of *lactobacillus*, *lactobacillus agilis*, *clostridia*, and *rikinellaceae* were higher in the lecithin treatment group, whereas the proportions of *prausnitzii, erysipelotrichi, lachnospiraceae,* and *alactolyticus* were higher in the control group after 21 days, which is the same as 42 days.

## 4. Discussion

### 4.1. Growth Performance and Organ Weight

The result of this trial revealed that direct addition of high purity lecithin into the diet of broilers showed no significant effects on the performance or relative organ weight throughout the whole experiment period, which is the same as most previous studies. The energy requirements of commercial broilers is very high, while the fat source is limited in traditional diet. Exogenous supplementation of lipid becomes an inevitable trend in broiler production for better performance. However, the effect of lipid or emulsifier additives remain inconsistent. Through exogenous supplementation of phospholipids, lecithin had different effects on performance improvement in several reports. Previous reports showed that supplementation of lecithin in low energy and protein diets improved both performance and digestibility parameters in broilers [19], while some reported the effect was not significant [11]. Meanwhile, a previous study showed that the body weight of groups supplemented with lecithin at 21 days and 35 days were not significantly different compared to the control group [20]. Additionally, even more, lecithin supplementation in animal diet showed suppression effect in gastric emptying and resulted in vomiting and diarrhea in some reports [21]. Meanwhile, lecithin was used as exogenous emulsifiers to improve the utilization of fat and energy in weaning piglets [22] or broilers [19,23]. Fats, such as hydrophobic components, must aggregate to form micelles to be absorbed. Emulsifiers found in the digestive tract (mainly bile salts) naturally mediate this process and improve the formation of micelles [24]. However, most of the studies reported that no significant influences were observed on the N or energy digestibility. Due to the short digestive tract and unstable digestibility of the broilers, the effect of lecithin in increasing the fat digestibility seems not obvious [25]. That may explain the non-significant effect of lecithin on growth performance or organ weight in broilers.

Overall, these results may be affected with the source of dietary lipids, the formation and addition amount of phospholipid product, the breed of the chickens and the duration of the trial. Improvement effect of lecithin on performance or organ weight may be more consistent and obvious in the low energy and protein diets in broilers.

### 4.2. Meat Quality

In this study, dietary supplementation with lecithin had a positive effect on meat color, water-holding capacity, and tenderness at different periods, which may be due to the effect of lecithin supplementation in lipid metabolism. Zhao et al. [10] reported that lecithin supplementation could increase the transportation of lipids in the body and improve the deposition of fat in the muscle, which is consistent with our results. Meat characteristics, including tenderness, water-holding capacity, pH, and meat color, are important indices for evaluation of meat quality. Meat color attribute for the final evaluation and acceptance of a meat product in the consumption [26]. In this study, the color value of breast meat was increased by lecithin supplementation. Similar results were reported and indicated that the color value of meat also showed a trend in improvement with the increase of emulsifier dosage [25,27]. As the most important textural characteristic of meat, tenderness has great impact on the consumer experience of broiler meat. Shear force value in this trail was significantly increased by lecithin treatment, which is also related to the fat content in the meat [28]. The pH value showed no difference in this trail, revealing that lecithin treatment showed no effect on lactic content.

### 4.3. Lipid Metabolism

The cholesterol level in serum of broilers after 21 days significantly increased with the lecithin supplementation. This trial provides evidence that serum cholesterol level of broilers remains constant at 42 days. Similar results were detected by Zhao et al. [10]. The reason may be the immature development of the digestive tract in chicks. Previous studies reported that lecithin supplementation could increase the duodenum development [29], and the effect is more obvious in the early stage. The results of those studies suggested that the effect of lecithin on the serum profile of broilers may be more efficient in the starter period [9]. Previous research on cholesterol metabolism using isotope tracer techniques indicated that the net balance of cholesterol homeostasis was relatively stable and resistant to plant sterol supplementation in chickens [30].

Exogenous emulsifier can accelerate the emulsification of lipids in the small intestine and promote the activation of lipase. In addition, lecithin was reported to promote the secretion of endogenous bile acid, and further improve the utilization rate of fat [25]. This trial showed that the enzyme activity of HL and GE was significantly increased by supplementation with lecithin at 21 days. No other parameters were significantly affected by dietary lecithin supplementation at 21 or 42 days. Previous research reported that there is a positive correlation between the enzyme activities of lipase and the deposition of lipids across both in broilers and layer strains [31]. As the main content of lipoprotein, lecithin plays an important role in the lipid metabolism such as lipid transportation. Exogenous phospholipids can elevate the HDL level and decrease the cholesterol level in serum, resulted in the regulation of lipid deposition [19,23]. The lipid metabolism results of this study indicated that the lipid transportation and deposition was altered by lecithin supplementation. Exogenous lecithin accelerated the lipodieresis in liver and reduced the transport of lipids in serum, which resulted in fewer lipid depositions in the abdomen. Boontiam et al. [32] reported that lecithin supplementation increase the proportion of lipids that used in muscle formation. That may neatly explain the results of meat quality in this experiment. As the visual indicators of lipid deposition in broilers, tenderness, and abdominal adipose were affected in this trial through the addition of lecithin, which validates the prediction that lipid metabolism is affected by the lecithin addition.

### 4.4. Microbial Community

The composition of intestinal microbiota is important for maintaining homeostasis of the gastrointestinal tract and the health of the host [33]. The intestinal microbial community has been recognized as a strong determining factor of host physiology, especially through its critical role in the digestion of feed in a host [34]. The 16S rDNA gene sequencing in cecal content shown that addition of lecithin altered the microbiota composition at the phylum level in broilers. PCA plots confirmed the results at the phylum level. In our study, *Bacteroidetes*, *Firmicutes*, *Tenericutes*, and *Proteobacteria* were the main bacteria found in the broiler intestinal flora, which was consistent with previous research [35]. More specifically, lower *Firmicutes* and higher *Bacteroidetes* levels were observed in the lecithin treatment groups, whereas higher levels of *Bacteroidetes* and *Proteobacteria* were observed in the control group. Studies comparing the gut microbiota between obese and lean animals showed that lower *Firmicutes* and higher *Bacteroidetes* levels were associated with the lean phenotype [36]. It can also be concluded from many studies that lower *Firmicutes* and higher *Bacteroidetes* levels lead to fewer lipid deposits in animals. *Lactobacillus* were observed as the bacterial genus which can cure enteritis, while the clostridia were reported as synergistic action with *lactobacillus* [37]. The higher distribution of *lactobacillus* and *Clostridia* in lecithin treatment group showed that the distribution of bacterial in the genus level may be helpful in the lipid absorption of birds. Therefore, the altered profiles of the cecal microbiota after 21 and 42 days was involved in the process of lecithin effect on broiler lipid metabolism.

## 5. Conclusions

In summary, by taking advantage of 16S rDNA sequencing, this study revealed that lower *Firmicutes* and higher *Bacteroidetes* in levels are noted in broiler ceca, responding to lecithin treatment. Lecithin supplementation improved the enzyme activity of HL and GE in serum, also reducing the abdominal adipose tissue in broilers. Finally, this study provides evidence that lecithin supplementation had a positive effect on meat color and tenderness in broilers.

## Figures and Tables

**Figure 1 animals-11-02537-f001:**
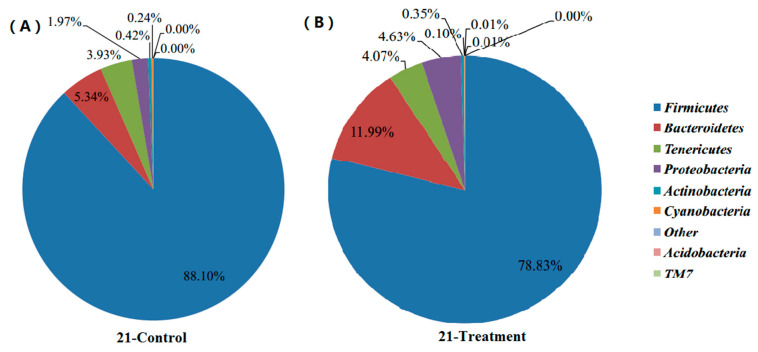
Effects of lecithin on the profiles of cecal microbial community after 21 days according to phylum for the control group (**A**) and lecithin treatment group (**B**).

**Figure 2 animals-11-02537-f002:**
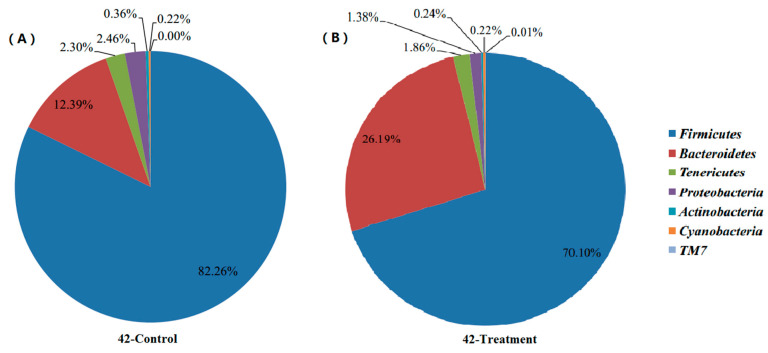
Effects of lecithin on the profiles of cecal microbial community after 42 days according to phylum for the control group (**A**) and lecithin treatment group (**B**).

**Figure 3 animals-11-02537-f003:**
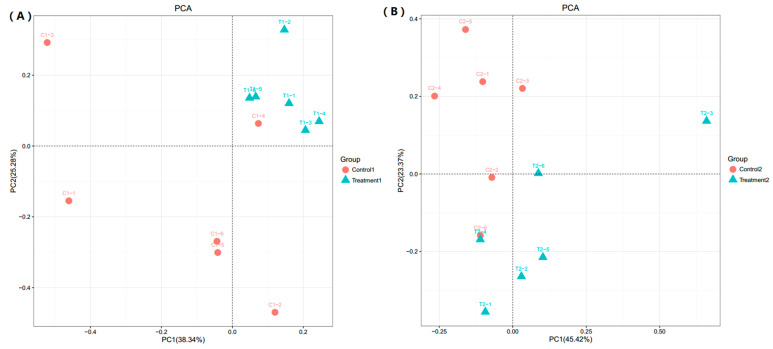
Principal components analysis (PCA) plots of samples from different days and groups: 21 (**A**) and 42 (**B**) days. Red dot means control group and blue dot means lecithin treatment group.

**Table 1 animals-11-02537-t001:** Diet composition and nutrient levels during the experiment (basal diets in dry basis).

Items	0–21 Days	22–42 Days
Ingredients (%)		
Corn	54.6	60.4
Soybean meal	35.2	30.2
Corn gluten meal	3.18	2.0
Soybean oil	2.65	3.52
Calcium hydrogen phosphate	2.0	1.65
Limestone	1.25	1.25
NaCl	0.35	0.35
Mineral premix ^1^	0.2	0.2
Vitamin premix ^2^	0.03	0.025
Aureomycin	0.03	0.03
Choline chloride (50%)	0.26	0.2
L-lysine (78%)	0.08	0.075
Methionine	0.17	0.1
Energy and nutrient composition		
Calculated values ^3^		
ME (Mcal/kg)	2.95	3.05
Crude protein (%)	22.28	19.78
Crude fat (%)	5.40	6.30
Ca (%)	1.17	1.05
Available phosphorus (%)	0.59	0.51
Lys (%)	1.18	1.04
Met (%)	0.50	0.40
Analyzed values		
Crude protein (%)	21.51	19.23
Crude fat (%)	5.23	6.16
Ca (%)	1	0.91
Available phosphorus (%)	0.46	0.4
Lys (%)	1.15	1.01
Met (%)	0.50	0.40

^1^ The mineral premix contents (mg/kg) were as follows: Fe, 80; Mn, 100; Cu, 8; Zn, 75; Se, 0.15; I, 0.35. ^2^ The vitamin premix provides the following per kg of diet: vitamin A, 12500 IU; vitamin D3, 2500 IU; vitamin E, 30 IU; Vitamin K3, 2.65 mg; vitamin B1, 2 mg; vitamin B2, 6 mg; vitamin B3, 50 mg; vitamin B5, 12 mg; vitamin B7, 0.0325 mg; vitamin B9, 1.25 mg; vitamin B12, 0.025 mg. ^3^ The values are calculated according to the values of feedstuffs in NRC (1994).

**Table 2 animals-11-02537-t002:** Effects of lecithin on the performance and mortality of broilers throughout the experiment ^1^.

Period	Items	Unit	Control	Treatment	SEM	*p-*Value
0–21 days	ADG	g	24.84	25.14	0.18	0.464
	ADI	g	39.48	39.39	0.33	0.898
	FCR	g/g	1.59	1.57	0.01	0.191
22–42 days	ADG	g	69.23	69.23	1.14	0.998
	ADI	g	143.50	139.45	1.48	0.180
	FCR	g/g	2.08	2.02	0.03	0.370
0–42 days	ADG	g	47.45	47.18	0.58	0.829
	ADI	g	84.72	84.88	0.59	0.901
	FCR	g/g	1.79	1.80	0.02	0.766
0–42 days	Mortality	%	3.13	1.56	0.007	0.438

^1^ These data are expressed as the mean. Pooled SEM are shown, the same as below. ADG means average daily gain, ADI means average daily intake, FCR means feed conversion ratio. Comparisons between the control and treatment groups based on a *t*-test. A *p* value less than 0.05 was considered to indicate statistical significance, the same as below.

**Table 3 animals-11-02537-t003:** Effects of lecithin on the relative organ weight ^1^ of broilers at 21 and 42 days.

Period	Items (%)	Control	Treatment	SEM	*p-*Value
21 days	Liver	2.65	2.40	0.11	0.265
	Heart	0.58	0.53	0.02	0.079
	Spleen	0.12	0.10	0.008	0.202
	Thymus	0.22	0.18	0.01	0.126
42 days	Liver	1.87	1.77	0.04	0.243
	Heart	0.43	0.44	0.01	0.462
	Spleen	0.12	0.12	0.006	0.920
	Thymus	0.24	0.23	0.02	0.734
	Abdominal adipose	1.58	1.00	0.13	0.012

^1^ Relative organ weight means the organ weight/BW*100%.

**Table 4 animals-11-02537-t004:** Effects of lecithin on the meat quality of broilers at 21 and 42 days.

Period	Items	Unit	Control	Treatment	SEM	*p-*Value
21 days	PH	-	5.59	5.58	0.03	0.843
	L*	-	50.53	53.98	0.79	0.023
	a*	-	11.97	13.43	0.22	0.078
	b*	-	4.47	4.09	0.41	0.406
	Drip loss	%	0.25	0.28	0.007	0.062
	Shear force	Kg/cm^2^	1.15	0.76	0.08	0.008
42 days	PH	-	5.49	5.35	0.06	0.269
	L*	-	52.11	51.49	0.55	0.589
	a*	-	11.97	13.43	0.30	0.052
	b*	-	3.75	2.26	0.52	0.008
	Drip loss	%	0.25	0.18	0.02	0.096
	Shear force	Kg/cm^2^	1.73	1.34	0.10	0.041

L* means relative lightness, a* means relative redness, and b* means relative yellowness.

**Table 5 animals-11-02537-t005:** Effects of lecithin on the serum biochemical parameters of broilers at 21 and 42 days ^1^.

Period	Items	Unit	Control	Treatment	SEM	*p-*Value
21 days	GLU	mmol/L	6.91	6.87	0.28	0.946
	CHO	mmol/L	3.88	3.48	0.08	0.012
	TG	mmol/L	0.40	0.44	0.01	0.129
	HDL	mmol/L	2.59	2.72	0.09	0.482
	LDL	U/L	0.78	0.84	0.03	0.267
42 days	GLU	mmol/L	5.57	6.92	0.24	0.002
	CHO	mmol/L	3.39	3.33	0.07	0.699
	TG	mmol/L	0.31	0.31	0.009	0.940
	HDL	mmol/L	2.56	2.31	0.08	0.134
	LDL	U/L	0.67	0.69	0.03	0.686

^1^ GLU means glucose, CHO means cholesterol, TG means triglyceride, LDL means low-density lipoprotein cholesterol, and HDL means high-density lipoprotein cholesterol.

**Table 6 animals-11-02537-t006:** Effects of lecithin on the enzyme activity in serum and liver of broilers at 21 and 42 days ^1^.

Period	Items	Unit	Control	Treatment	SEM	*p-*Value
21 days(serum)	LPL	U/mL	1.53	1.64	0.07	0.419
	HL	U/mL	2.34	1.53	0.16	0.008
	GE	U/mL	3.96	3.18	0.17	0.018
21 days (liver)	HL	U/mL	0.48	0.55	0.03	0.228
42 days(serum)	LPL	U/mL	1.17	1.77	0.02	0.176
	HL	U/mL	1.69	1.96	0.11	0.213
	GE	U/mL	2.87	3.78	0.24	0.054

^1^ LPL means lipoprotein lipase and HL means hepatic lipase. GE means general esterase activity, which was calculated as the sum of LPL and HL.

**Table 7 animals-11-02537-t007:** Genus level significance analysis of cecal microbial community after 21 days.

Genus	Control	Treatment	SEM	*p*-Value
*Prausnitzii*	8666.75	5636.88	2178.65	<0.01
*Lactobacillus*	3352.38	3592.38	1006.36	0.04
*Lactobacillus agilis*	292.00	652.63	397.56	<0.01
*Erysipelotrichi*	1285.75	807.50	298.67	0.08
*Lachnospiraceae*	1236.38	771.88	210.19	0.03
*Clostridia*	1256.00	277.57	198.28	<0.01
*Alactolyticus*	1880.13	642.75	4699.79	<0.01
*Rikenellaceae*	5007.00	5922.50	1487.29	0.06
*Closttridiales*	1857.14	224.43	329.16	<0.01

**Table 8 animals-11-02537-t008:** Genus level significance analysis of cecal microbial community after 42 days.

Genus	Control	Treatment	SEM	*p*-Value
*Prausnitzii*	6513.25	4190.88	1295.28	<0.01
*Lactobacillus*	2379.63	3292.75	512.86	0.03
*Lactobacillus agilis*	4622.00	7488.00	3278.18	<0.01
*Erysipelotrichi*	822.63	750.50	172.89	0.45
*Lachnospiraceae*	1024.25	680.63	116.82	<0.01
*Clostridia*	457.25	2957.13	1729.83	<0.01
*Alactolyticus*	158.50	286.38	42.18	0.53
*Rikenellaceae*	5638.38	1704.75	768.61	<0.01
*Closttridiales*	1241.88	1172.88	196.28	<0.01

## Data Availability

Data are available if requested.

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
