# Peer review of "Evaluation of a Lecithin Supplementation on Growth Performance, Meat Quality, Lipid Metabolism, and Cecum Microbiota of Broilers"

_animals, 2021, doi:10.3390/ani11092537_

Round 1
Reviewer 1 Report
The article is easy to follow and a common subject to me. It was very nice to read it. I did some notifications in the file attached. Please verify it. Congratulations on your research.

Author Response
Response to Reviewer 1 Comments
Date: August , 2021
Evaluation of a lecithin supplementation on growth performance, meat quality, lipid metabolism and cecum microbiota of broilers
Dear Reviewer,
Thank you very much for your hard and kind work on our paper (Manuscript ID: animals-1327052). We feel lucky that our manuscript went to you as the valuable comments from you not only helped us with the improvement of our manuscript, but suggested some neat ideas for further studies. We have studied comments carefully and have made correction that we hope meet with approval.
We agree with your suggestion and revised this in the manuscript. All the changes are marked in red in both this response letter and the revised manuscript.
Thanks for your patience in reading this letter. We hope that our responses and revisions are acceptable. Please do not hesitate to contact me if more revisions are required.
Yours sincerely,
Yiru Shen
Poultry Institute, Chinese Academy of Agricultural Sciences
Yangzhou 225125, Jiangsu
P. R. China
Email: shenyiru929@126.com

Reviewer 2 Report
This a simple study with just two treatments and there is some new information (microbial profile) - but there are also major weaknesses.
In particular, there are TWO flaws.
First, the BW of broilers at 42d is less than 2 kg. This is much lower than the BW obtained under commercial operations and questions the scientific and practical validity of the data.
Second, no measurement is made on lipid digestibility and energy utilisation – given that these are the reasons for the use of lecithin emulsifier. Thus the relevance of the data is questionable. The product can change microflora, but without effects on lipid utilisation there is little practical use. Authors have missed an important point in the conduct.
INTRODUCTION is weak with some incorrect statements (L65, 71, 73). Need be revised.
Table 1: formatting needs complete revision. See poultry publications. Analysed values must be provided, especially for crude fat. The focus of this study is lipids and even calculated fat value are missing. Why NRC (1994) recommendations are used for Arbor Acres- over 25 years old and not appropriate for modern broilers !!
Methodology details must be expanded . for example, drip loss – after what time duration.
TABLES: Pooled SEM (not SE of means) must be provided. Use the footnotes to present more details (no of observations etc.). improve table legends. Follow decimal convention (e.g. 8666, not 8066.75 etc.). Check units. Avoid the term organ index – use relative organ weight, % BW. What is the importance of lowering serum lipids in chickens?
DISCUSSION is poor and wanting. L300 – why?. L315-324: confusing discussion and incoherent
Poor language throughout – too many deficiencies – listing all issues will not be possible.
Author Response
Response to Reviewer 2 Comments
Date: August , 2021
Evaluation of a lecithin supplementation on growth performance, meat quality, lipid metabolism and cecum microbiota of broilers
Dear Reviewer,
Thank you very much for your hard and kind work on our paper (Manuscript ID: animals-1327052). We feel lucky that our manuscript went to you as the valuable comments from you not only helped us with the improvement of our manuscript, but suggested some neat ideas for further studies. We have studied comments carefully and have made correction that we hope meet with approval.
Please find below details regarding all the changes we have made to the manuscript in response to your comments. All the changes are marked in red in both this response letter and the revised manuscript.
Point 1: The BW of broilers at 42d is less than 2 kg. This is much lower than the BW obtained under commercial operations and questions the scientific and practical validity of the data.
Response 1: Thank you for your insightful suggestion. We have double check our data. There is no miscalculation.
The main cause of the lower BW in this trail is the feed type. As described in Line 90, the diet of the birds was provided in mash form to avoid the component destruction of lecithin. As we can concluded from many studies published in recent years, the weight gain of broilers fed in mash(powder) diet that about 2 kg is common and acceptable. Different test conditions would produce different results.
Reference 1: Qorbanpour M, Fahim T, Javandel F, Nosrati M, Paz E, Seidavi A, Ragni M, Laudadio V, Tufarelli V. Effect of Dietary Ginger (Zingiber officinale Roscoe) and Multi-Strain Probiotic on Growth and Carcass Traits, Blood Biochemistry, Immune Responses and Intestinal Microflora in Broiler Chickens. Animals (Basel). 2018 Jul 14;8(7):117. doi: 10.3390/ani8070117.
Reference 2: Osman A, Bin-Jumah M, Abd El-Hack ME, Elaraby G, Swelum AA, Taha AE, Sitohy M, Allam AA, Ashour EA. Dietary supplementation of soybean glycinin can alter the growth, carcass traits, blood biochemical indices, and meat quality of broilers. Poult Sci. 2020 Feb;99(2):820-828. doi: 10.1016/j.psj.2019.12.026. Epub 2020 Jan 24.
Point 2:No measurement is made on lipid digestibility and energy utilisation – given that these are the reasons for the use of lecithin emulsifier. Thus the relevance of the data is questionable. The product can change microflora, but without effects on lipid utilisation there is little practical use. Authors have missed an important point in the conduct.
Response 2:Thank you for your insightful suggestion. We agree with your suggestion.
Lecithin was used as exogenous emulsifiers to improve the utilization of fat and energy in animals. However, most of the studies reported that no significant influences were observed on the N or energy digestibility. Due to the short digestive tract and unstable digestibility of the broilers, the effect of lecithin in increasing the fat digestibility seems not obvious as well. According to the inconsistent effect of lecithin as emulsifiers , we did not measure the lipid digestibility and energy utilization. On the other hand, the lipid digestibility ties closely with fat transportation and disposition. Related parameter were measured and discussed in the manuscript.
Considering that readers would have the same doubt, we added the discussions in Line 1002-1028.
Point 3: INTRODUCTION is weak with some incorrect statements (L65, 71, 73). Need be revised.
Response 3:Thank you for your insightful suggestion. We agree with your suggestion.
We have revised this in Line 60-81.
Point 4:Table 1: formatting needs complete revision. See poultry publications. Analysed values must be provided, especially for crude fat.
Response 4: Thank you for your insightful suggestion. We agree with your suggestion which help our presentation to be more canonical.
We have revised this in Table 1. Calculated value and analyzed value of the diet are added.
Point 5:Methodology details must be expanded . for example, drip loss – after what time duration.
Response 5:Thank you for your insightful suggestion. We agree with your suggestion.
We have revised this in Line 175-204. Detailed methodology and reference are added.
Point 6:TABLES: Pooled SEM (not SE of means) must be provided. Use the footnotes to present more details (no of observations etc.). improve table legends. Follow decimal convention (e.g. 8666, not 8066.75 etc.). Check units. Avoid the term organ index – use relative organ weight, % BW. What is the importance of lowering serum lipids in chickens?
Response 6:Thank you for your insightful suggestion. We agree with your suggestion.
We have revised this in the manuscript.
According to Helkin et al., blood lipid parameter are considered as key factors of lipid metabolism balance. The findings in this trail are consistent with recent studies, which demonstrate modifications in blood lipid profiles of broiler chickens with the dietary lecithin supplementation.
Reference 3: Helkin A, Stein JJ, Lin S, et al. (2016) Dyslipidemia part 1 – review of lipid metabolism and vascular cell physiology. Vasc Endovascular Surg 50, 107–118.
Point 7: DISCUSSION is poor and wanting. L300 – why?. L315-324:confusing discussion and incoherent.
Response 7:Thank you for your insightful suggestion. We agree with your suggestion.
We have revised this in Line 1219-1031. Different parts are divided. Some errors are corrected.
Point 8: Poor language throughout.
Response 8:Thank you for your insightful suggestion. We agree with your suggestion.
We have revised this in the manuscript.
Thanks for your patience in reading this long letter. We hope that our responses and revisions are acceptable. Please do not hesitate to contact me if more revisions are required.
Yours sincerely,
Yiru Shen
Poultry Institute, Chinese Academy of Agricultural Sciences
Yangzhou 225125, Jiangsu
P. R. China
Email: shenyiru929@126.com

Round 2
Reviewer 2 Report
Revision is acceptable